# Porphyrin as a versatile visible-light-activatable organic/metal hybrid photoremovable protecting group

Adiki Raja Sekhar [1], Youhei Chitose[2,3,4], Jiří Janoš [5], Sahar Israeli Dangoor [6], Andrea Ramundo [2,3], Ronit Satchi-Fainaro [6], Petr Slavíček [5], Petr Klán [2,3 ✉] & Roy Weinstain [1 ✉]

Photoremovable protecting groups (PPGs) represent one of the main contemporary implementations of photochemistry in diverse fields of research and practical applications. For the past half century, organic and metal-complex PPGs were considered mutually exclusive classes, each of which provided unique sets of physical and chemical properties thanks to their distinctive structures. Here, we introduce the *meso*-methylporphyrin group as a prototype hybrid-class PPG that unites traditionally exclusive elements of organic and metal-complex PPGs within a single structure. We show that the porphyrin scaffold allows extensive modularity by functional separation of the metal-binding chromophore and up to four sites of leaving group release. The insertion of metal ions can be used to tune their spectroscopic, photochemical, and biological properties. We provide a detailed description of the photoreaction mechanism studied by steady-state and transient absorption spectroscopies and quantum-chemical calculations. Our approach applied herein could facilitate access to a hitherto untapped chemical space of potential PPG scaffolds.

[1] School of Plant Sciences and Food Security, Faculty of Life Sciences, Tel-Aviv University, Tel-Aviv 6997801, Israel. [2] Department of Chemistry, Faculty of Science, Masaryk University, Kamenice 5, 625 00 Brno, Czech Republic. [3] RECETOX, Faculty of Science, Masaryk University, Kamenice 5, 625 00 Brno, Czech Republic. [4] Basic Chemistry Program, Graduate School of Advanced Science and Engineering, Hiroshima University, Higashi-Hiroshima 739-8526, Japan. [5] Department of Physical Chemistry, University of Chemistry and Technology, Prague, Technická 5, 16628 Prague 6, Czech Republic. [6] Department of Physiology and Pharmacology, Sackler School of Medicine, Tel Aviv University, 6997801 Tel Aviv, Israel. ✉email: klan@sci.muni.cz; royweinstain@tauex.tau.ac.il

**P**hotoremovable protecting groups (PPGs), also known as photocaged compounds, are linked to a molecule of interest to mask its chemical or biological activity. Upon irradiation at a specific wavelength, the PPGs are tracelessly removed, allowing non-invasive spatio-temporal control over the release of free molecules with a high degree of chemoselectivity in complex chemical and biological environments. They represent one of the main contemporary applications of photochemistry in diverse research areas, ranging from chemistry and material science to biology and medicine[1–4].

Since their introduction more than 50 years ago[5,6], dozens of PPG types have been developed[7,8]. Their fundamental, mutually exclusive classification differentiates between organic and transition-metal complex PPGs (Fig. 1a). Transition-metal-free (organic) PPGs possess an organic chromophore and liberate molecules via the cleavage of a covalent bond[7,8]. This large class of PPGs encompasses a wide range of molecular scaffolds, from the established nitroaryl and coumarin groups to the more recent, visible-light absorbing BODIPY and cyanine groups[7,8]. In contrast, metal-complex PPGs consist of a central metal atom or ion surrounded by ligands as part of their chromophores. The complexes have unique ground- and excited-state properties arising from the nature of the central metal, the ligands, and specific interactions between them. The metal coordination center serves as a *bonafide* reaction center in all metal-complex PPGs reported to date, and ligand exchange is the ultimate liberation step, succeeding a photoinitiated metal-to-ligand, ligand-to-metal, or ligand-to-ligand charge-transfer transition[7–10]. Notable examples in this class include cob(III)alamins, and ruthenium and rhodium complexes[9,10].

Over the decades, the conceptual dichotomy between these two PPG classes has been maintained, being translated into essentially non-converging development paths. To date, efforts to exploit the distinctive properties of organic and metal-complex photo-activatable systems within a single structure have primarily focused on either linking the established metal-free PPGs to metal complexes, metal surfaces, or metal nanoparticles or, on the other hand, on the conjugation of metal-complex PPGs to organic chromophores (antennas)[8]. We considered a hybrid class of PPGs, which allows (but does not require) the incorporation of a metal (ion) as a part of the chromophore and releases leaving groups through the photoinduced cleavage of covalent bonds (Fig. 1b). We reasoned that blurring the traditional dividing line between these two PPG classes would provide access to a yet untapped chemical space of this new type of PPGs.

In this work, to evaluate the concept, we repurpose the ubiquitous ligand porphyrin as an independent PPG. We show that the functional separation of the metal-binding chromophore and the site of leaving group release can be leveraged to fine-tune its spectroscopic and photochemical properties by simply introducing metal ions into the porphyrin core. The release of leaving groups from the free-ligand and metal-containing porphyrin derivatives is studied by both steady-state and transient-absorption spectroscopies, and the results are supported by quantum-chemical calculations. We provide a detailed description of the photoreaction mechanism, laying the foundations for further expansion and optimization of such hybrid-class PPGs.

## Results and discussion

**Metal-free porphyrin PPGs**. The design of a porphyrin-based PPG was inspired by the recent emergence of the *meso*-methyl functionalized boron dipyrromethene (BODIPY) group[11,12]. Hückel molecular orbital predictions showed that BODIPY PPGs could release a leaving group thanks to the negative charge increase at the *meso* position upon excitation[13]. The structurally related porphyrin scaffold has four such positions, which exhibit a similar electronic behavior upon excitation to the first excited state (see Supplementary Section 4).

The synthesis of an appropriate porphyrin core was accomplished by MacDonald acid-catalyzed condensation (Fig. 2a)[14]. The *ortho*-unsubstituted phenyl group of **1** and the aliphatic aldehyde in **2** facilitated a scrambling mechanism[15,16] that gave rise to the formation of all possible porphyrin combinations (Supplementary Fig. 1), including the key A₃B-type derivatives **3**. Subsequent alcohol deprotection provided the metal-free *meso*-

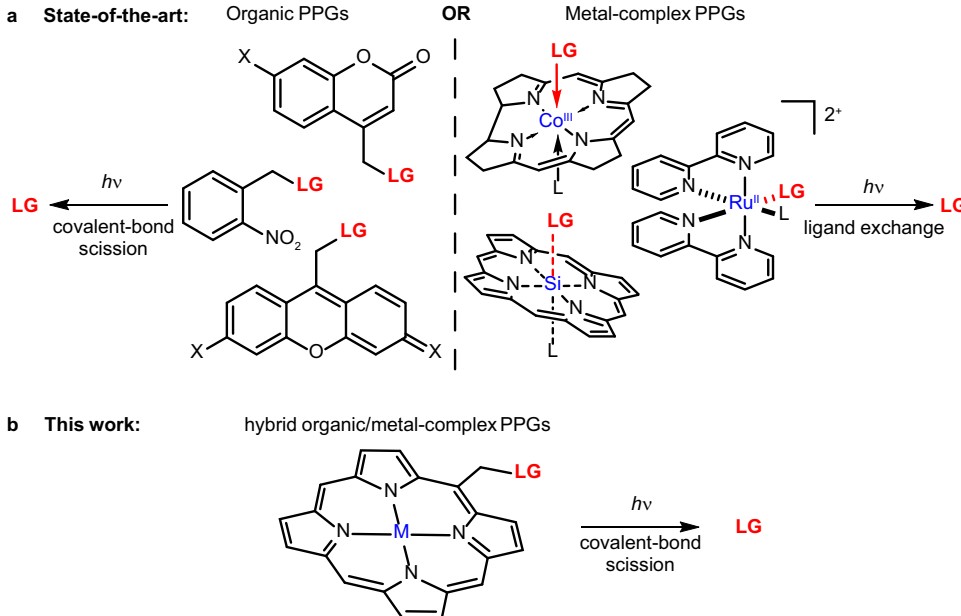

**Fig. 1 Meso-methylporphyrin as a hybrid organic/metal-complex PPG scaffold. a** Contemporary organic and metal-complex PPGs; the photorelease of leaving groups (LG) occurs via a covalent bond scission or ligand exchange process, respectively. **b** *Meso*-methylporphyrin as a hybrid organic/metal-complex PPG: diverse metal ions can be incorporated into the chromophore to modulate its properties, and a leaving group is photoreleased via covalent bond scission.

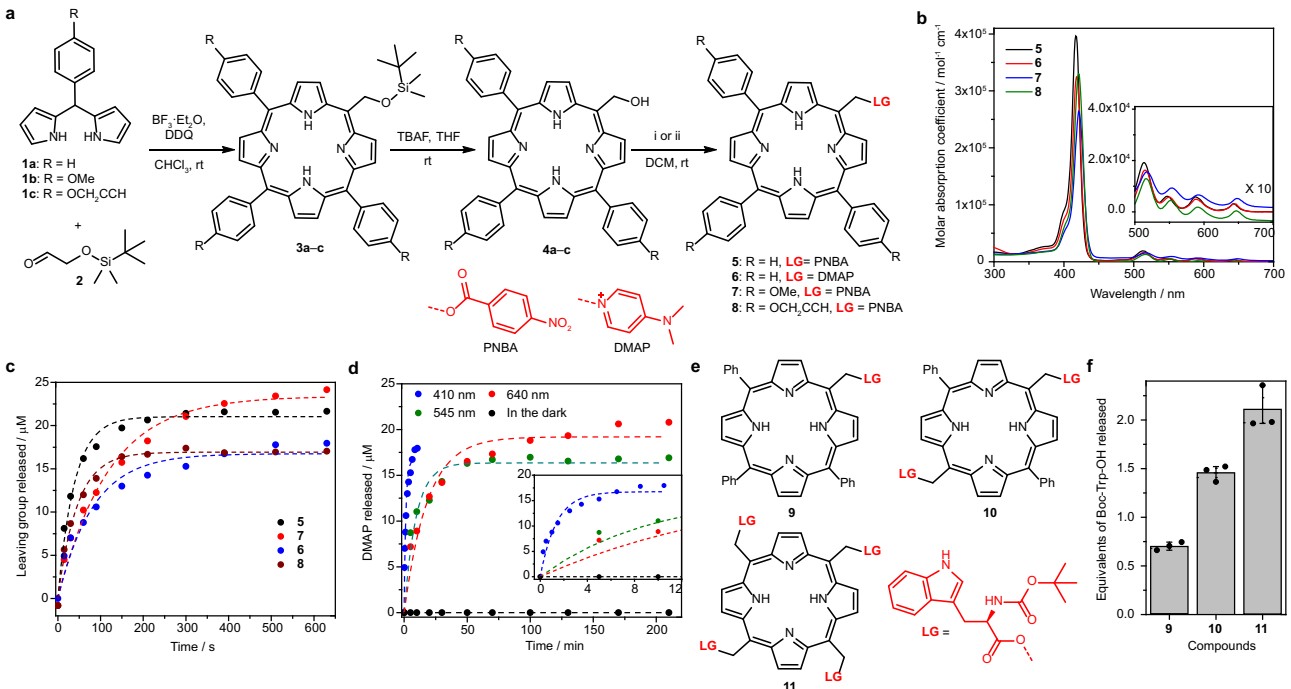

**Fig. 2 *Meso*-methylporphyrin as a versatile PPG scaffold: synthesis and photochemistry. a** Synthesis of *meso*-methylporphyrin derivatives. i: PNBA, DMAP, DCC, 20 °C. ii: mesylchloride, DMAP, 20 °C. **b** Absorption spectra of PNBA- and DMAP-caged *meso*-methylporphyrins (3 μM, aerated DMSO). Inlet: a 10-fold magnification of the Q-band regions. **c** Photorelease of leaving groups from *meso*-methylporphyrin derivatives **5**–**8** (25 μM, degassed DMSO, $\lambda_{irr}$ = 410 nm, 40 mW cm$^{-2}$; HPLC-MS analysis). **d** Photolysis of **6** (25 μM, degassed DMSO) at $\lambda_{irr}$ = 410 nm (18 mW cm$^{-2}$); 545 nm (52 mW cm$^{-2}$); and 640 nm (67 mW cm$^{-2}$). Inlet: focus on early time points. **e** Structures of mono-, di- and tetra-$N_\alpha$-Boc-(L)-Trp caged *meso*-methylporphyrin derivatives (**9**–**11**). **f** Quantification of the photorelease of leaving groups from **9**–**11** ($\lambda_{irr}$ = 410 nm, 40 mW cm$^{-2}$, HPLC-MS analyses). Averages from three experiments are shown; error bars represent the standard deviation.

methylporphyrin **4**. The hydroxyl group in **4** is a poor nucleophile; its reactions with chloroformate and isocyanate were not successful. However, it was found to be sufficiently nucleophilic to react with *O*-acylisourea derivatives or mesyl chloride, allowing its in-situ displacement with nucleophiles such as *p*-nitrobenzoic acid (PNBA) or *N,N*-dimethylaminopyridine (DMAP) to give the final derivatives **5** and **6**, respectively.

The photophysical and photochemical properties of **5** and **6** in DMSO or methanol solutions are summarized in Table 1. Analogous to metal-free porphyrins[17,18], the S$_0$ → S$_2$/S$_1$ electronic transitions (Q bands) were found in the range of 500–650 nm, and very intense absorption bands with the maxima $\lambda_{max}^{abs}$ in the blue region were assigned to the S$_0$ → S$_4$/S$_3$ transitions (Soret bands; Fig. 2b and Supplementary Fig. 2)[19]. The photorelease of leaving groups (LGs) from **5** and **6** was monitored by an LG appearance (HPLC) and starting material disappearance (absorption spectroscopy; Fig. 2c and Supplementary Fig. 3). The LGs were liberated with chemical yields of >70% and photodecomposition quantum yields ($\Phi_{dec}$) of ~0.002 upon irradiation at 410 nm. It is worth noting that the large molar absorption coefficients at the irradiation wavelength ($\lambda_{irr}$) are responsible for very high uncaging cross sections ($\Phi_{dec} \varepsilon_{max} (\lambda_{irr})$)[7,8] of >650 M$^{-1}$ cm$^{-1}$.

The structural and physico-chemical properties of porphyrin allow for a high degree of modularity in its application as a PPG. For example, the photorelease of an LG from **6** can be achieved by irradiation to its different absorption bands in the visible region (Fig. 2d and Supplementary Fig. 4). The fluorescence quantum yield ($\Phi_F$; in the range of 420–560 nm) was found to be wavelength-independent (Supplementary Table 1). In addition, some structural modifications of the basic scaffold can be implemented without compromising its functionality. Substituents on the *meso*-phenyl groups generally modulate the electronic

properties of the porphyrin chromophore only to a small extent[20]. Indeed, derivatives **7** and **8**, featuring the 4-methoxyphenyl or 4-*O*-propargylphenyl groups (Fig. 2a), exhibited spectroscopic and photochemical properties similar to those of **5** (Table 1). The lower $\Phi_{dec}$ found for **8** can be explained by the increased rotational freedom of the molecule[21]. Nevertheless, the alkyne groups in **8** provided convenient reaction sites for further functionalization via a copper-click reaction (Supplementary Fig. 5). An additional level of modularity emerges from the multiple *meso* positions in the porphyrin scaffold. Taking advantage of this structural feature, A$_3$B-, A$_2$B$_2$- and B$_4$-type porphyrin PPGs bearing $N_\alpha$-Boc-*L*-tryptophan as a model LG were synthesized (**9**–**11**, Fig. 2e). The number of equivalents of an LG released was twofold from the A$_2$B$_2$-type **10** and 2.9-fold from the B$_4$-type **11** when compared to that from the A$_3$B-type **9** (Fig. 2f), probably involving an enhanced non-productive photodegradation of the chromophore during the sequential release of multiple LGs.

To evaluate the functionality of the *meso*-methylporphyrin group in a complex environment, we synthesized derivatives bearing the tubulin assembly inhibitor indibulin or the antifolate agent methotrexate (MTX) as LGs (**12** and **13**, respectively, Fig. 3a, b). Compound **12** released indibulin upon irradiation (Fig. 3c), but it was found to be thermally unstable in a PBS buffer (Supplementary Fig. 6), unlike a recently reported coumarin-caged indibulin that was effectively applied in cultured cells[22]. Conversely, **13** was thermally stable in a solution, and its photoactivation was successfully conducted upon 545 nm irradiation because MTX itself absorbs and decomposes when photolyzed at 410 nm (Fig. 3c and Supplementary Fig. 6). Thus, **13** was used for a cellular evaluation in cultures of the murine 4T1 mammary carcinoma cell line. Fluorescence microscopy indicated

**Table 1 Photophysical and Photochemical Properties of *meso*-Methylporphyrin Derivatives.**

| | Absorption[a] | Emission[a] | | | Photoreaction[b] | | | |
|---|---|---|---|---|---|---|---|---|
| | $\lambda_{abs}$/nm ($\varepsilon$/10$^4$ M$^{-1}$ cm$^{-1}$) | $\lambda_{max}^F$/nm | $\Phi_F$ | $\tau_F$/ns | yield/%[c] | $\Phi_{dec}$[d] | $\Phi_{dec}\varepsilon$[e] | $\Phi_\Delta$[f] |
| 5 | 417 (39.1), 513 (1.9), 547 (0.6), 588 (0.6), 643 (0.3) | 648, 715 | 0.045 ± 0.002 | 9.41 ± 0.01 | 86 | 0.0017 ± 0.0002 | 664 | 0.721 ± 0.002[g] |
| 6 | 418 (32.8), 515 (1.6), 549 (0.6), 586 (0.5), 644 (0.3) | 647, 716 | 0.082 ± 0.001 | 11.12 ± 0.02 | 72 | 0.0023 ± 0.0003 | 747 | 0.159 ± 0.007 |
| 7 | 422 (26.1), 518 (1.5), 553 (0.9), 593 (0.6), 648 (0.5) | 654, 720 | 0.049 ± 0.001 | 5.71 ± 0.01 | 96 | 0.0022 ± 0.0005 | 573 | 0.728 ± 0.029[g] |
| 8 | 421 (32.8), 516 (1.6), 553 (0.7), 591 (0.5), 647 (0.4) | 653, 719 | 0.041 ± 0.001 | 6.52 ± 0.01 | 71 | 0.0012 ± 0.0001 | 394 | 0.714 ± 0.022[g] |
| 12 | 420 (46.2), 515 (2.0), 550 (0.7), 589 (0.6), 645 (0.2) | 648, 715 | 0.076 ± 0.002 | 11.19 ± 0.02 | 64 | 0.0017 ± 0.0003 | 784 | 0.154 ± 0.009 |
| 13 | 417 (44.3), 513 (1.9), 548 (0.6), 590 (0.6), 644 (0.4) | 647, 714 | 0.063 ± 0.001 | 10.38 ± 0.02 | 71 | 0.0021 ± 0.0003 | 930 | 0.158 ± 0.003 |
| 6-Zn | 427 (37.2), 560 (1.5), 602 (0.5) | 607, 661 | 0.037 ± 0.002 | 2.44 ± 0.01 | 74 | 0.0256 ± 0.0006 | 9533 | 0.245 ± 0.002 |
| 6-Pd | 417 (22.3), 525 (1.9), 556 (0.5) | 647, 716 | 0.003 ± 0.001 | ND | 74 | 0.0004 | 102 | 0.231 ± 0.002 |
| 6-Cu | 420 (23.8), 545 (1.2) | ND | ND | ND | 0 | 0.0 | 0 | 0.014 ± 0.002 |
| 6-Ni | 417 (19.6), 531 (1.4) | ND | ND | ND | 0 | 0.0 | 0 | 0.007 ± 0.002 |

[a] Solutions in aerated DMSO, $c = 3 \times 10^{-6}$ M.
[b] Photoreaction parameters were measured in degassed DMSO except for quantum yields of the singlet oxygen formation ($\Phi_\Delta$).
[c] Chemical yield of the release.
[d] Determined by irradiation at 410 nm for 6, 6-Zn, 6-Pd, 6-Cu, and 6-Ni using ferrioxalate as an actinometer; the remaining values were calculated with 6 as a reference. [e] Units of M$^{-1}$ cm$^{-1}$.
Quantum yields of the singlet oxygen formation were measured in: [f] methanol using flavonol as a reference ($\Phi_\Delta = 0.070$) or [g] benzene using tetraphenylporphyrin (TPP) as a reference ($\Phi_\Delta = 0.66$). ND = not detected.

that the molecule is taken up into cells and appears to be localized in punctate structures that do not co-localize with lysosomal compartments (Fig. 3d, e). The maximum intracellular signal was observed after 2-h preincubation (Supplementary Fig. 7a, b). Cell viability measurements (using a Coulter counter) demonstrated that compound **5**, bearing a non-toxic LG, has low light-dependent toxicity (Fig. 3f), which was attributed to the photodynamic effect (in accord with $\Phi_\Delta$; Table 1). In comparison, **13** showed a much lower IC$_{50}$ when exposed to 545 or 640 nm light (IC$_{50}$ = 2.6 and 2.8 μM, respectively), the values of which are similar to or lower than that of free MTX at the same concentration. Collectively, the above results confirm the viability of *meso*-methylporphyrin as a versatile PPG scaffold.

**Metal-containing porphyrin PPGs.** The ability of porphyrin to chelate a large number of metal cations provides an unprecedented approach to manipulating the properties of this PPG. We synthesized complexes of **6** chelating metal cations with distinct electronic configurations (i.e., Zn(II), Pd(II), Cu(II), or Ni(II) in compounds **6-Zn**, **6-Pd**, **6-Cu**, and **6-Ni** respectively; Fig. 4a). A single-crystal X-ray diffraction analysis of **6-Zn** is shown in Supplementary Fig. 8 (Supplementary Table 2). The photophysical and photochemical properties of metal porphyrins are summarized in Table 1. The metal-complexes **6-Pd**, **6-Cu**, and **6-Ni** exhibited $\lambda_{max}^{abs}$ values similar to that of **6** but with lower molar absorption coefficients ($\varepsilon_{max}$) and fluorescence quantum yields ($\Phi_F$, Fig. 4b and Supplementary Fig. 2). In comparison, derivative **6-Zn** possessed a 9-nm red-shifted $\lambda_{max}^{abs}$ value, a slightly higher $\varepsilon_{max}$, and a shorter fluorescence lifetime. Upon irradiation at 410 nm in DMSO solutions, the release of DMAP was observed from **6-Zn** and **6-Pd** in 74% chemical yields. However, no photorelease was observed for derivatives **6-Cu** and **6-Ni** (Fig. 4c and Supplementary Fig. 3). The photodecomposition quantum yield $\Phi_{dec}$ of **6-Zn** was more than an order of magnitude higher than that of the free-ligand derivative **6** (Table 1), resulting in an extraordinarily high uncaging cross section of $\Phi_{dec} \varepsilon_{max} = 9533\,\text{M}^{-1}\,\text{cm}^{-1}$, one of the highest to be reported to date[7,8]. The photodecomposition of **6-Pd** was ~6-times less efficient than that of **6**.

We also examined an excitation wavelength ($\lambda_{exc}$) dependence of fluorescence and photorelease efficiencies. Analogous to compound **6**, $\Phi_F$ was found to be practically $\lambda_{exc}$ independent for **6-Zn** in the range of 420–560 nm (Supplementary Table 1). More importantly, the quantum yield values of DMAP release ($\Phi_r$; HPLC) from **6-Zn** were found within a narrow range of 0.016, 0.008, and 0.007 when irradiated at Soret and Q bands at $\lambda_{irr}$ = 427, 560, and 600 nm, respectively (Supplementary Table 3); therefore, the entire visible range can be used for excitation. The $\Phi_r$ values are lower than those of $\Phi_{dec}$ because the maximum chemical yield of LG release reached 74% (i.e., 26% of excited molecules are decomposed unproductively; Table 1).

**Mechanistic studies.** A detailed mechanistic study was carried out to understand the photochemistry of LG photorelease from *meso*-methylporphyrin derivatives and support the further development of these hybrid-class PPGs.

*Photoproducts.* A solvent-capture photoproduct **4a** was formed upon irradiation of **6** in degassed DMSO/water, along with porphyrin-*meso*-carbaldehyde **14** and other unidentified photoproducts (Fig. 5a and Supplementary Fig. 9). The formation of **4a** pointed either to a *meso*-methyl cation intermediacy, analogous to that observed in *meso*-methyl BODIPY PPGs[23], or to an S$_N$2-like process. Aldehyde **14** was most probably formed upon oxidation by DMSO (Supplementary Fig. 9c), as previously reported in some coumarin PPGs[24]. Significantly higher amounts of the solvent-capture

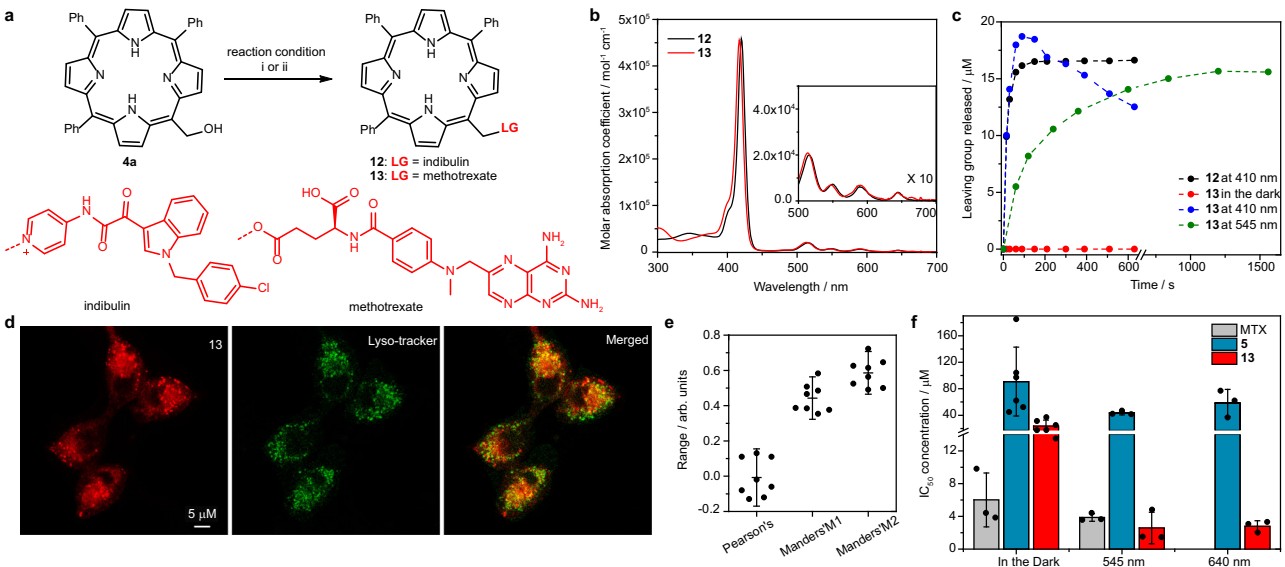

**Fig. 3 *Meso*-methylporphyrin PPG is functional in cellular environment. a** Synthesis of *meso*-methylporphyrin derivatives bearing anti-cancer drugs. i: mesylchloride, indibulin, 20 °C. ii: methotrexate, DMAP, DCC, 20 °C. **b** Absorption spectra of anti-cancer-caged *meso*-methylporphyrins (3 μM, aerated DMSO). Inlet: a 10-fold magnification of the Q-band regions. **c** Photorelease of leaving groups from *meso*-methylporphyrin derivatives **12** and **13** (25 μM, degassed DMSO, $\lambda_{irr}$ = 410 nm, 40 mW cm$^{-2}$ or 545 nm 52 mW cm$^{-2}$; HPLC-MS analysis). **d** Representative fluorescence microscopy images of the cellular distribution of **13** in cultured murine 4T1 mammary carcinoma cells. Red: compound **13**, green: LysoTracker Green. **e** Co-localization parameters of **13** and LysoTracker Green. The middle line represents the mean, and error bars represent the standard deviation. **f** Calculated IC$_{50}$ concentrations of **5** and **13** in cultured murine 4T1 mammary carcinoma cells following or not irradiation with 545 nm (5 min) or 640 nm (10 min) light. The experiment was repeated at least three times in triplicates; error bars represent the standard deviation.

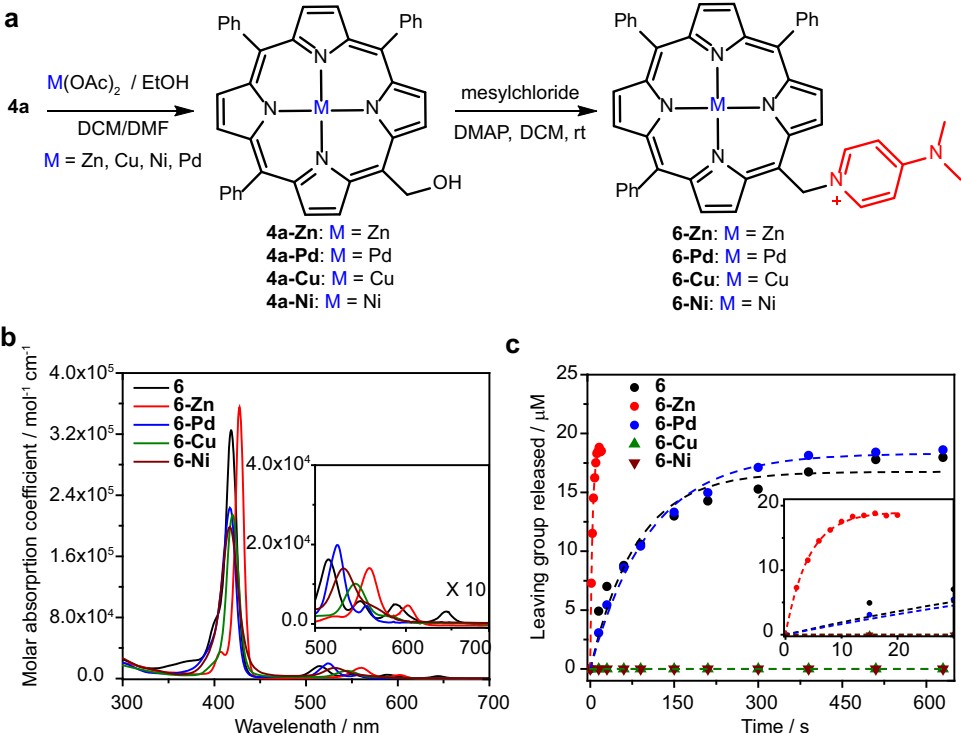

**Fig. 4 Metal-containing *meso*-methylporphyrin PPGs: synthesis and photochemistry. a** Synthesis of metal-containing *meso*-methylporphyrins followed by the installation of DMAP as a model leaving group. **b** Absorption spectra of DMAP-caged metal-containing *meso*-methylporphyrin derivatives (3 μM, aerated DMSO); that of an analogous metal-free *meso*-methylporphyrin (**6**) is shown for comparison. Inlet: a 10-fold magnification of the Q-bands region. **c** Photorelease of DMAP from metal-containing *meso*-methylporphyrin derivatives (25 μM, degassed DMSO, $\lambda_{irr}$ = 410 nm, 40 mW cm$^{-2}$; HPLC-MS analysis), the reaction of an analogous metal-free *meso*-methylporphyrin (**6**) is shown for comparison. Inlet: focus on early time points.

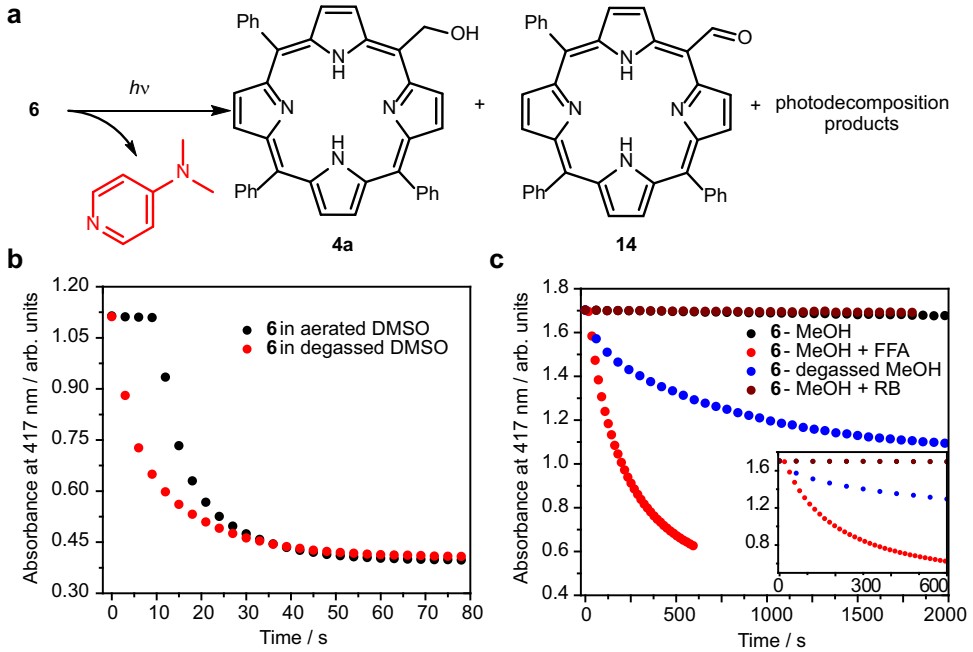

**Fig. 5 Photoreaction kinetics of DMAP-caged porphyrin derivatives. a** Photoproducts formed upon photoexcitation ($\lambda_{irr}$ = 410 nm, 18 mW cm$^{-2}$) of **6** (25 μM) in a degassed DMSO solution. **b** Photochemistry of **6** (3 μM) in aerated or degassed DMSO solutions. **c** Photochemistry of **6** (3 μM) in aerated or degassed methanol solutions ($\lambda_{irr}$ = 420 nm) and in the presence of a singlet oxygen generator (rose bengal, RB, $\lambda_{irr}$ = 545 nm, 52 mW cm$^{-2}$) or quencher (furfuryl alcohol, FFA). Inlet: focus on early time points.

photoproduct compared to that of **6** were found upon irradiation of **6-Zn** in degassed DMSO/water or methanol solutions (Supplementary Fig. 10), implying a profound effect of the metal type on the porphyrin PPG photochemistry[25,26].

*Effects of oxygen.* Monitoring the photodegradation of **6** and **6-Zn** at short time intervals and low-intensity light (410 nm, 3.2 mW cm$^{-2}$) revealed a delay in the onset of photolysis when performed in aerated compared to degassed DMSO or DMSO/water solutions (Fig. 5b and Supplementary Fig. 11). In contrast, the photodegradation of **6** in methanol was observed only in the absence of oxygen (Fig. 5c). In addition, the photoreaction of **6** in aerated methanol proceeded with the same efficiency in the presence of a singlet oxygen quencher (furfuryl alcohol, Fig. 5c and Supplementary Fig. 12), which excluded singlet oxygen as a photorelease mediator. These results suggest that the initial $^1O_2$ generation by an excited porphyrin outcompetes the photorelease, which is in accord with the measured quantum yields of both processes ($\Phi_\Delta$ is ~2 orders of magnitude higher than $\Phi_{dec}$, Table 1). The unusual photoreaction course observed in aerated DMSO solutions thus most probably reflects the initial quenching of the excited porphyrin by oxygen to form singlet oxygen, which is depleted (removed) from a solution in the presence of DMSO molecules[27], and then the photorelease can proceed efficiently. This competing process is not available in methanol. A similar effect of local molecular oxygen depletion on a photoreaction has recently been reported for the photoactivation of gold(I) aryl ethynyl complexes phosphorescence[28]. The ability of ground-state oxygen to quench the photorelease process also suggests that the reaction proceeds from the triplet excited state.

The singlet oxygen production efficiencies of $\Phi_\Delta$ = 0.16, 0.25, and 0.23, determined for **6**, **6-Zn**, and **6-Pd**, respectively, in methanol, and relatively low fluorescence quantum yields measured for these compounds (Table 1) were used to estimate the upper limit of ISC quantum yield (<0.7). The relatively low values of $\Phi_{dec}$ (<0.02) are approximately 2 orders of magnitude

lower than those of $\Phi_\Delta$; therefore, the production of singlet oxygen is the major competing pathway to the LG release. Fortunately, singlet oxygen is not primarily responsible for the chemical degradation of the PPGs in our work (for example, the excitation of a singlet oxygen generator, rose bengal, at $\lambda_{irr}$ = 545 nm in the presence of **6** in aerated methanol did not lead to its decomposition, Fig. 5c).

*Transient spectroscopy.* The excited-state radiative and non-radiative deactivation pathways for metal-free and metal-containing model porphyrin derivatives (tetrakis(4-carboxyphenyl)porphyrin, see Supplementary Section 3), which do not bear an LG, and for porphyrin PPGs, **6**, **6-Zn**, **6-Pd**, and **6-Cu**, were studied by femtosecond (fs) and nanosecond (ns) transient absorption (TA) spectroscopies. We found that neither the absence of an LG in the *meso*-methyl position nor the presence of 4-carboxyl groups on *meso*-phenyl substituents in the model derivatives had any significant effect on the spectroscopic behavior of all studied compounds.

The fs-TA spectra, evolution-associated spectrum (EAS) components, and their relative populations for porphyrins **6**, **6-Zn**, **6-Pd**, and **6-Cu** in DMSO are shown in Supplementary Fig. 13, and the data are summarized in Table 2. Global analyses identified four evolution-associated spectra (EAS) components for **6** and **6-Zn**. As in the case of the model porphyrins, we can attribute the observed components (EAS1, EAS2) to the transitions occurring within Q bands ($k_Q^1$ and $k_Q^2$, Table 2), such as vibrational cooling or conformational relaxation. For example, it has been reported that conformational relaxation in tetraphenylporphyrin occurs in the ~100 ps time scale[29]. Analogous to the kinetics of the model compounds (see Supplementary Section 3) and also other porphyrin derivatives[30–35], the transition between EAS3 and EAS4 was attributed to singlet–triplet intersystem crossing (ISC; $k_{ISC} = 1.7 \times 10^8$ s$^{-1}$ and $2.2 \times 10^8$ s$^{-1}$ for **6** and **6-Zn**, respectively). A much faster ISC ($k_{ISC} = 1.6 \times 10^{10}$ s$^{-1}$) and only two EAS components were

**Table 2 Major Photophysical and Photochemical Processes Observed in 6, 6-Zn, 6-Pd, and 6-Cu[a].**

| | $k_Q^1$/s$^{-1}$ | $k_Q^2$/s$^{-1}$ | $k_{ISC}$/s$^{-1}$ | $k_d$/s$^{-1}$ | | | | $\Phi_F$ | $\tau_F$/ns |
|---|---|---|---|---|---|---|---|---|---|
| | | | | DMSO | | Methanol | | | |
| | | | | Aerated | Degassed | Aerated | Degassed | | |
| **6** | $4.1 \times 10^{10}$ | $3.7 \times 10^{9}$ | $1.7 \times 10^{8}$ | $6.3 \times 10^{5}$ | $9.1 \times 10^{4}$ | $3.3 \times 10^{6}$ | $1.7 \times 10^{5}$ | 0.082 | 11.12 |
| **6-Zn** | $6.9 \times 10^{10}$ | $8.0 \times 10^{9}$ | $2.2 \times 10^{8}$ | $4.0 \times 10^{5}$ | $6.0 \times 10^{4}$ | $2.1 \times 10^{6}$ | $9.4 \times 10^{4}$ | 0.037 | 2.44 |
| **6-Pd** | ND | ND | $1.6 \times 10^{10}$ | $6.7 \times 10^{5}$ | $1.1 \times 10^{5}$ | $3.2 \times 10^{6}$ | $6.6 \times 10^{5}$ | 0.0034 | ND |
| **6-Cu** | ND | ND | ND | $2.9 \times 10^{10}$ | | | | ND | ND |

[a]fs-TA spectroscopy: $\lambda_{exc} = 387$ nm; concentrations -0.2–0.4 mM. ns-TA spectroscopy: $\lambda_{exc} = 355$ nm; concentrations 6–18 µM. $k_Q^1$ and $k_Q^2$—the rate constants of transitions within the Q states, $k_{ISC}$—intersystem crossing, $k_d$—triplet-state decay, $\Phi_F$—fluorescence quantum yield, $\tau_F$—fluorescence lifetime. ND = not determined. Emission properties were recorded in DMSO.

observed for **6-Pd**, which is related to the heavy-atom effect of the Pd atom.

The fs-TA spectra of a Cu(II) derivative **6-Cu** displayed a narrow band at ~440 nm in the ps time scale (EAS1), but the rapid subsequent processes made it difficult to monitor all evolution steps. A weak EAS1 → EAS2 transition ($k = 6.7 \times 10^{11}$ s$^{-1}$) was followed by the appearance of a very weak transient signal in the region of 440–520 nm (EAS3), and it was accompanied by an artifact monitored at 530–700 nm, attributed to the signal of the solvated electron[36]. The decay rate constant of the EAS2 → EAS3 transition was determined to be $k = 2.9 \times 10^{10}$ s$^{-1}$ (Table 2). This observed kinetics is analogous to that of the model Cu-porphyrin derivative discussed in Supplementary Section 3. The previous interpretations of analogous data assigned the first two transients as a tripdoublet $^2$T state, a $^2$[d,d] exciplex state, formed by the ligation with O-coordinating solvents, respectively[36–38]. We conclude that the uncaging process from **6-Cu** cannot compete with a very fast radiationless decay of the productive excited state.

The ns-TA spectra of **6**, **6-Zn**, and **6-Pd** in aerated and degassed DMSO or methanol are shown in Supplementary Fig. 14, and the data are summarized in Table 2. The triplet lifetimes of these compounds were found to be ~6-times shorter in aerated than degassed DMSO due to quenching of the triplet state by oxygen. Shorter triplet lifetimes in **6-Pd** are related to a faster triplet–singlet ISC (heavy atom effect)[30,31,34,35]. Very similar kinetic data were obtained in methanol solutions (Table 2); all rate constants were only slightly faster than those obtained in DMSO. It is worth mentioning that the triplet lifetimes of porphyrin PPGs show a clear correlation with their photodecomposition quantum yields ($\Phi_{dec}$), further indicating that the photorelease of an LG proceeds via the triplet excited state.

**Quantum chemical calculations and the proposed release mechanism.** Nucleophilic substitution of a leaving group, calculated on simplified meso-methylporphyrin model compounds bearing an LG (Supplementary Section 4), was found to be a slightly endergonic process for the reaction with water as a nucleophile under standard conditions (Supplementary Table 6). Based on the experimental observations discussed in detail above and ab initio calculations and simulations, we propose the photoreaction mechanism involving several key steps: the formation of a triplet excited molecule from the singlet excited state via intersystem crossing, the release of an LG to give a carbocation in the triplet state, and its intersystem crossing and the reaction with a nucleophile (Fig. 6).

Porphyrin derivatives can be excited into a highly absorbing Soret band. This state typically represents a third or fourth excited state in free-base or Zn porphyrins[19,39–41], as was also simulated on model free-base, Zn, and Pd porphyrins using the TDDFT/BMK, CIS, and ROCIS methods (Supplementary Table 7). In contrast, model Cu porphyrin possesses a large number of weakly

absorbing states below the Soret bands (Supplementary Table 7b). Our dynamical simulations on free-base, Zn, and Pd porphyrins, the models for **6**, **6-Zn**, and **6-Pd** derivatives, predict an ultrafast internal conversion, completed in ~100 fs (Supplementary Fig. 20). These porphyrins in the $S_1$ state deactivate primarily via intersystem crossing (Supplementary Table 8). Indeed, the importance of coupling between vibrations and electronic transitions has previously been emphasized[40]. Based on our calculations, we propose that the leaving group as an anion is liberated from the triplet state hereby formed via an $S_N1$-like mechanism, which is the rate-limiting step for the overall release process. The calculated activation Gibbs energies for the formation of the corresponding carbocation in the triplet state, strongly stabilized by resonance, varied in the range of 0.8–1 eV (Supplementary Table 9), which corresponds to the rate constants of up to $10^5$ s$^{-1}$. A long triplet lifetime is critical for the release process and controls its efficiency. Note that the energy barriers are similar for the cleavage from the $S_1$ electronic state, which, however, has a much shorter lifetime, detrimental to the release. The situation is different for **6-Cu**, where the metal d-electrons strongly interact with those of the porphyrin ring, resulting in a very efficient excited-state deactivation (tens of picoseconds)[36,37,42]. This fast process is usually interpreted in the literature as IC to the $^2S_1$ state (the $S_1$ state of porphyrin with a doublet state of Cu$^{2+}$), accompanied by fast ISC to the $^2T_1$ state, where a longer kinetic component is assigned to the formation of ligand–porphyrin exciplex or the formation of dark charge-transfer $^2T_1$ states[36,37,42]. Based on our ab initio calculations, we suggest another possible deactivation pathway; porphyrin **6-Cu** has a much lower $^2S_0/^2S_1$ gap, as the $^2S_1$ state is located well below the Q band, and, therefore, direct IC is possible as an alternative deactivation channel (Supplementary Table 7b). This view is strongly supported by our experimental transient-absorption data (see Table 2 and, for model Cu-porphyrin systems, also Supplementary Fig. 17). A shorter lifetime of the EAS1 → EAS2 transition (see above) is then interpreted as IC from the Soret band states to the $^2S_1$ state with no ISC involved and a longer lifetime (EAS2 → EAS3) would correspond to IC from $^2S_1$ to the ground state.

In conclusion, the meso-methylporphyrin derivatives introduced in this work possess several favorable properties for a PPG, including structural modularity, strong absorbance of light in the visible range of the spectrum, and high photouncaging efficiencies. The capacity to generate up to four productive photocleavable sites within a single molecule should enable the concomitant release of different leaving groups and could, for example, be valuable in the construction of light-responsive polymer networks or hydrogels[43–47]. The relatively high efficiency of $^1O_2$ generation exhibited by the porphyrin derivatives reported herein might be a limiting factor in some biological applications, where the photodynamic effect is detrimental, although several

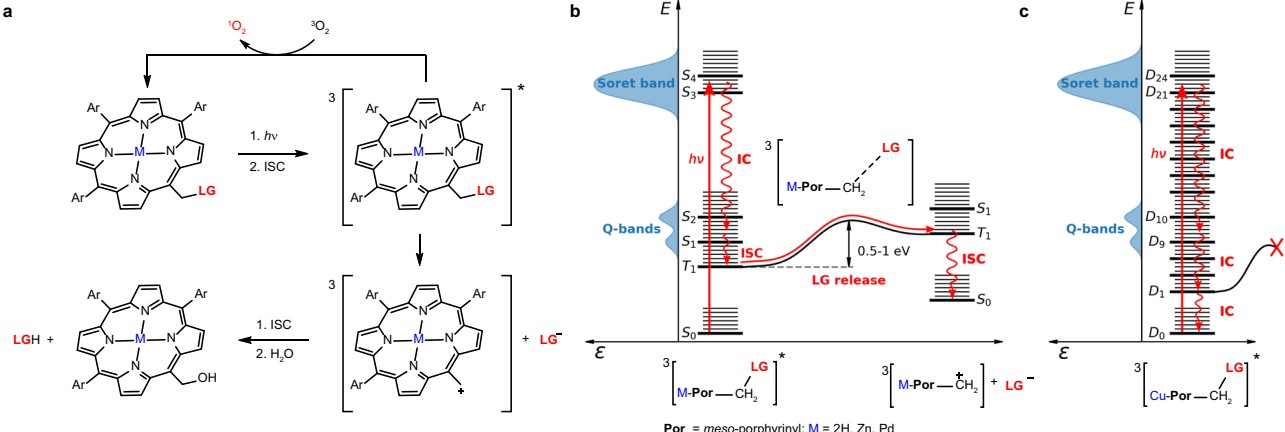

**Fig. 6 Photochemistry of DMAP-caged porphyrin derivatives. a** The proposed photorelease mechanism for porphyrin PPGs. **b** An extended Jablonski diagram for free-base, Zn, and Pd porphyrin PPGs. Intensities of the UV/visible transitions are also depicted. **c** An extended Jablonski diagram for a Cu porphyrin PPG.

mitigation strategies are known[48–52]. On the other hand, efficient $^1O_2$ generation can be synergistic to caging cytotoxic drugs[4,53].

The *meso*-methylporphyrin group represents a novel prototype of a PPG that unites traditionally exclusive properties of organic and metal-complex PPGs within a single structure. The functional separation between the metal-binding chromophore and the site of leaving group release allows for an elective introduction of metal ions to manipulate its photophysical and photochemical properties. We expect that this approach could be implemented to repurpose additional known metal-binding ligands as PPGs, opening thus a new chemical space for the design of photocaged compounds with unique structural, chemical and photochemical properties.

## Methods

**General materials and methods**. Reagents and solvents were purchased from Sigma-Aldrich, Combi-Blocks, or TCI and used as received unless otherwise stated. Anhydrous solvents and reagents (dichloromethane (DCM), tetrahydrofuran (THF), dimethylsulfoxide (DMSO), and dimethylformamide (DMF)) were obtained as Sure Seal bottles from Sigma-Aldrich, Alfa Aesar, or ACROS Organics™, dried using the standard methods when necessary. Thin-layer chromatography and flash chromatography were performed using EMD pre-coated silica gel 60 $F_{254}$ plates and silica gel 60 (230–400 mesh), respectively. Preparative HPLCs were performed on a Waters HPLC chromatograph with an XBridge C18 column (100 × 19 mm, 5 μm) with water containing TFA (0.1%, solvent A) and acetonitrile containing TFA (0.1%, solvent B) with the flow rate of 15 mL min⁻¹. For model compounds, HPLC-UV/Vis analyses were performed on an Agilent 1260 Infinity II HPLC system with a SIELC Obelisc N column (150 × 4.6 mm, 5 μm) using a water (0.1% TFA)/acetonitrile gradient of 10% to 100% of solvent B at a flow rate of 2 mL min⁻¹.

$^1$H and $^{13}$C NMR spectra were collected in CDCl₃ or DMSO-$d_6$ (Cambridge Isotope Laboratories, Cambridge, MA) at 400 and 101 MHz, respectively, at 25 °C using a Bruker Advance III spectrometer at the Department of Chemistry NMR Facility at Tel-Aviv University or on 300 and 125 MHz spectrometers at the Department of Organic Chemistry at Masaryk University, Brno. All chemical shifts are reported in the standard δ notation of parts per million (ppm) relative to either the tetramethylsilane signal (TMS, δ = 0.00 ppm) or residual solvent peak as an internal reference. $^{13}$C NMR chemical shifts are reported in ppm relative to the CDCl₃ or DMSO-$d_6$ signal as internal standards. HPLC-MS analysis was performed on a Waters HPLC SYNAPT system with an XBridge C18 column (100 × 3 mm, 5 μm) using a water (0.1% of TFA)/acetonitrile (0.1% of TFA) gradient from 0 to 100% of solvent B at the flow rate of 1 mL min⁻¹. Low-resolution ESI mass spectrometry was performed on an LC/MS Acquity QDa detector coupled with the Waters HPLC.

**Absorption and emission spectroscopy**. UV–vis absorbance spectra were recorded on an Agilent Cary 60 UV–vis spectrophotometer with 1 cm quartz cells in aerated DMSO at room temperature for all synthetic porphyrins. Molar absorption coefficients (ε) and absorbance maxima ($\lambda_{max}$) were determined in DMSO using Beer's law from the plots of absorbance vs. concentration in 1 cm quartz cells in DMSO. Fluorescence spectra were recorded on an automated

luminescence spectrometer in 1 cm quartz fluorescence cuvettes at 25 ± 1 °C; the sample concentration was set to keep the absorbance below 0.1 at $\lambda_{max}$; each sample was measured three times, and the spectra were averaged. Fluorescence quantum yields were determined using an integration sphere as the absolute values. For each sample, the fluorescence quantum yield was measured five times, and the results were averaged. Time-resolved fluorescence decays were determined using a nanosecond hydrogen arc lamp as the light source and time-correlated single-photon counting as the detection method. The fluorescence lifetimes were calculated from the first-order exponential decay fitting.

**Monitoring photolysis by absorption spectroscopy**. A solution of the given porphyrin (3 μM in 3 mL of DMSO or methanol or DMSO/PBS (1:1), either degassed (argon purged) or aerated solvent system) was placed in a 1 cm PTFE screw-cap fluorescence quartz cuvette equipped with an internal magnetic stirrer. The sample was irradiated by a Prizmatix light source (410/40 Mic-LED, UHP-T-White equipped with 545/30, or 640/40 nm filter) or by a 420/30 nm 32 LEDs light source (lab-made). Full absorption spectra of the sample were recorded at the time intervals specified for each experiment using a diode-array spectrophotometer in a kinetic mode. The reaction progress was monitored as a change in the absorbance at $\lambda_{max}$ of the respective porphyrin. Light intensity at the center of the cuvette was measured by a light meter (OPHIR photonics) in all experiments. Where specified, the singlet oxygen quencher (furfuryl alcohol, FFA, 150 μM) or the singlet oxygen generator (rose bengal, RB, 9 μM) were added to the reaction mixture before irradiation.

**Monitoring photolysis by HPLC analysis**. A solution of the given porphyrin (25 μM in 3 mL of DMSO or methanol or DMSO/PBS (1:1), either degassed (argon purged) or aerated) in a 1 cm path length PTFE screw-cap fluorescence quartz cuvette equipped with an internal magnetic stirrer was irradiated with a Prizmatix light source (410/40 Mic-LED, UHP-T-White equipped with 545/30, or 640/40 nm filter). Decomposition of the porphyrin PPGs and release of the leaving group was quantitated by HPLC-MS analyses (XBridge C18 column (100 × 3 mm, 5 μm) using water (0.1% of TFA)/acetonitrile (0.1% of TFA) gradient of 0–100% of solvent B at a flow rate of 1 mL/min). To monitor and quantify DMAP release, an Obelisc N column (150 × 4.6 mm, 100 Å, 5 μm) was used on a similar HPLC-MS setup. Calibration curves for all caged reference compounds and photoproducts were generated in the range of 5–150 μM.

**Photodecomposition quantum yields ($\Phi_{dec}$)**. Photodecomposition quantum yields for porphyrin PPGs were determined at $\lambda_{irr}$ = 410 nm (LEDs or a mercury-xenon arc lamp) in degassed DMSO solutions (freeze-pump-thaw technique; 3 cycles) using ferrioxalate as an actinometer. The reaction conversions were always kept below 10% to avoid the interference of photoproducts. The data were processed using single value decomposition (SVD), assuming the first-order rate law. The amount of absorbed light was calculated from the whole emission spectrum of the light source. All quantum yield measurements were repeated at least 3 times with independently prepared samples.

The rate of porphyrin decomposition can be expressed as

$$v_r = -\frac{d[\text{Por}]}{dt} = \frac{dn(\text{Por})}{Vdt} \tag{1}$$

where $v_r$ is the rate of decomposition, [Por] is the concentration of a porphyrin derivative, $n$(Por) is the amount of a porphyrin derivative in moles, and $V$ is the sample volume. The photodecomposition quantum yield $\Phi_{dec}$ was calculated

according to:

$$\Phi_{\text{dec}} = \frac{dn(\text{Por})}{q_{n,p}^0 [1 - 10^{-A(\lambda)}]dt} \quad (2)$$

where $\Phi_{\text{dec}}$ is the photodecomposition quantum yield, $n(\text{Por})$ is the amount of a porphyrin derivative in moles, $q_{n,p}^0$ is the spectral photon flux of incident photons ($m^{-2} s^{-1}$), $A(\lambda)$ is the absorbance of the sample at the given wavelength.

Combining the two equations gives:

$$\Phi_{\text{dec}} = \frac{v_r[\text{Por}]V}{q_{n,p}^0 [1 - 10^{-A(\lambda)}]} \quad (3)$$

**Determination of quantum yields of the singlet oxygen formation ($\Phi_\Delta$).** The singlet oxygen formation quantum yields ($\Phi_\Delta$) of all porphyrins were determined in either methanol or aerated benzene as a solvent with 9,10-dimethylanthracene (DMA) as a singlet oxygen quencher using flavonol or tetraphenylporphyrin (TPP)[54,55] as a standard photosensitizer. The singlet oxygen formation quantum yield is expressed as[56]

$$\Phi_\Delta \frac{I_{\text{abs}}}{k} = \Phi_\Delta^{\text{ref}} \frac{I_{\text{abs}}^{\text{ref}}}{k^{\text{ref}}} \quad (4)$$

where "ref" denotes the reference compound, $k$ denotes the first-order rate constant of the DMA consumption determined from the decrease in the absorption peak at 400 nm, and $I_{\text{abs}}$ is the absorption of excitation light by the photosensitizers (PS), which is dependent on the concentration, the molar absorption coefficient, and the intensity of the incident light. The relationship can be described by:

$$I_{\text{abs}} = \int I_{\text{laser}}(\lambda)(1 - e^{-\varepsilon(\lambda)cl})d\lambda \quad (5)$$

where $I_{\text{laser}}(\lambda)$ is the normalized emission spectrum of the 410 nm laser, $\varepsilon(\lambda)$ is the absorption coefficient, $c$ is the concentration of the dye, and $l$ is the path length of the fluorescence quartz cuvette.

**Transient absorption spectroscopy.** The nanosecond laser flash photolysis setup was generally operated in a right-angle arrangement of the pump and probe beams. Laser pulses of ≤700 ps duration at 355 nm (160 mJ) were obtained from an Nd:YAG laser. The laser beam was dispersed onto a 40-mm long and 10-mm wide modified fluorescence cuvette held in a laying arrangement. An over pulsed Xe arc lamp was used as the probe light source. Kinetic decay traces were recorded using a photomultiplier. Transient absorption spectra were obtained using an ICCD camera equipped with a spectrograph. Samples were degassed by three freeze-pump-thaw cycles or nitrogen bubbling for at least 15 min.

Femtosecond transient absorption was measured with the pump–super continuum probe technique using a Ti/Sa laser system (775 nm, pulse energy of 0.9 mJ, full width at half-maximum of 150 fs, operating frequency of 426 Hz). One part of the beam was fed into a non-collinear optical parametric amplifier (NOPA). The output at 387 nm was frequency-doubled by a β-barium borate (BBO) crystal to $\lambda = 266$ nm and, upon compression, elicited pump pulses of 1 μJ energy and <150 fs pulse width. The probe beam was generated by focusing the 775 nm beam in front of a CaF$_2$ plate with a 2 mm path length that produced a supercontinuum spanning a wavelength range of 270–690 nm. The pump and probe beams were focused to a 0.2 mm spot on the sample that was flowing in an optical cell with a thickness of 0.4 mm. The probe beam and a reference signal obtained by passing the solution next to the pump beam were spectrally dispersed and registered with two photodiode arrays (512 pixels). Transient absorption spectra were calculated from the ratio of the two beams. The pump–probe cross-correlation was <100 fs over the entire spectrum. The measurements were performed at ambient temperature (20 ± 2 °C). The data of the fs TA spectrum was collected until 3.1 ns. The global analysis was performed to get evolution-associated spectra (EAS). Each kinetic rate constant for the rise and decay of an EAS component was determined from the fitting result. Femtosecond TA spectra of **6**, **6-Zn**, **6-Pd**, and **6-Cu** derivatives were measured in DMSO. Nanosecond TA spectra measurements were carried out in DMSO and methanol using the 355-nm excitation for **6**, **6-Zn**, and **6-Pd**.

**X-ray crystallography.** Data were collected on a Bruker KAPPA APEX II diffractometer equipped with an APEX II CCD detector using a TRIUMPH monochromator with a Mo K$_\alpha$ X-ray source ($\alpha = 0.710, 73$ Å). The crystals were mounted on a cryoloop with Paratone oil, and all data were collected at 293 K. Crystal structures were solved using direct methods followed by full-matrix least-squares refinements against F$^2$ (all data are in HKLF 4 format) using SHELXTL[57]. Non-hydrogen atoms were refined with independent anisotropic displacement parameters, and hydrogen atoms were placed geometrically and refined using the riding model. Structure solution, refinement, graphics, and creation of publication materials were performed on SHELXL 97, PLATON 99[58], and WinGX system Ver-1.6414.

**Stability in the dark.** Stock solutions of all porphyrin compounds (DMSO, ~2 mM) were stored in the dark at 4 °C for at least 9 months and were found to be chemically stable by LC/MS analysis.

**Cell culture materials and methods.** L-Glutamine, penicillin, streptomycin, nystatin, sodium pyruvate, HEPES buffer, Dulbecco's Phosphate-Buffered Saline (PBS), Trypsin EDTA, and mycoplasma detection kit were purchased from Biological Industries Ltd. (Kibbutz Beit HaEmek, Israel). Fetal bovine serum (FBS) and Roswell Park Memorial Institute (RPMI) 1640 medium were purchased from Gibco-Thermo Fisher (Waltham, MA, USA). D-(+)-Glucose was purchased from Sigma-Aldrich (Rehovot, Israel). Green LysoTracker was purchased from Thermo Fisher Scientific.

**Cell culture.** Murine mammary adenocarcinoma 4T1 cells were purchased from American Tissue Culture Collection (ATCC, Manassas, VA, USA). 4T1 cells were cultured in RPMI 1640 medium supplemented with FBS (10%), penicillin (100 U mL$^{-1}$), streptomycin (100 μg mL$^{-1}$), nystatin (12.5 U mL$^{-1}$), L-glutamine (2 mM), glucose 45% (42 mM), sodium pyruvate (1 mM), and HEPES buffer (10 mM). Cells were routinely tested for mycoplasma contamination with a mycoplasma detection kit. Cells were grown at 37 °C, under CO$_2$ (5%).

**Cytotoxicity assays.** 4T1 cells were seeded on 6-well plates ($5 \times 10^3$ cells per well) and allowed to attach overnight. Cells were then incubated with the indicated concentrations of **5** or **13** in DMSO dissolved in a cell culture medium for 3 h. Following incubation, wells were washed twice with PBS, and a fresh medium was added. Cells were irradiated at 545 nm for 5 min or 640 nm for 10 min or kept in the dark. Following irradiation, cells were kept at 37 °C, under CO$_2$ (5%) in the dark for 72 h. Following incubation, cells were washed with PBS, detached by trypsin EDTA and counted by a Coulter counter (Beckman Coulter, Brea CA, USA). The concentration required to inhibit 50% cell survival in vitro is represented by the inhibitory concentration IC$_{50}$.

**Cellular distribution.** 4T1 cells were seeded on glass-bottom 24-well plates (Cellvis, California, USA; $5 \times 10^3$ cells per well) and allowed to attach overnight. Cells were then incubated with 100 μM of **13** and dissolved in a cell culture medium for 3 h. Cells were then washed with PBS and incubated with 75 nM of green LysoTracker for 30 min. Following incubation, wells were washed twice with PBS, and fresh PBS was added for imaging. Cells were imaged with a laser scanning confocal microscope (Zeiss LSM 780 inverted microscope) equipped with a 63×/1.35 Oil objective lens. Compound **13** was excited with the 405 nm laser, and the emitted light was collected between 580 and 690 nm. LysoTracker Green was excited by a 488 nm laser, and the emission was collected at 499–526 nm.

The colocalization plugin Coloc2 for the ImageJ (Fiji) software was used to generate 2D histograms and to determine Pearson correlation coefficients as well as Manders overlap coefficients. Non-corrected and non-thresholded images of **13** and LysoTracer were loaded into the plugin. Mean Pearson correlation coefficients for **13** and LysoTracer pair were determined from the entire field of view ($n \geq 6$) for several images.

**Cellular uptake kinetics.** 4T1 cells were seeded on glass-bottom 24-well plates (Cellvis, California, USA; $5 \times 10^3$ cells per well) and allowed to attach overnight. Cells were then incubated with 100 μM of **13** dissolved in a cell culture medium for 1–3 h. Following incubation, wells were washed twice with PBS, and fresh PBS was added for imaging. Cells were imaged with a laser scanning confocal microscope (Zeiss LSM 780 inverted microscope) equipped with a 63×/1.35 Oil objective lens. Compound **13** was excited with a 405 nm laser, and the emitted light was collected between 580 and 690 nm. Image analysis and signal quantification were done using the measurement function in the ZEN lite2012 software.

**Statistical analysis.** One-way ANOVA with Tukey correction or student $t$-test were performed in the Origin software, and statistical significance was determined at $p < 0.05$.

**Gibbs free energy for photolysis reaction.** The Gibbs free energies were evaluated via a thermodynamic cycle involving gas-phase energetics and solvation energies. First, the geometries were optimized, and the free energy was calculated in the gas phase. Then, the gas phase energies were corrected with solvation energies in DMSO calculated with a polarizable continuum model (PCM). The solvation energies of the proton were taken from the literature[59]. The energies were calculated with the Gaussian09 suite of programs, revision D.01[60].

**Calculations of excited electronic states.** The excitation energies and oscillator strengths at the time-dependent density functional (TDDFT) with Boese-Martin for Kinetics (BMK) functional were calculated with the 6–31+g* basis set for all structures except for the structure with **Pd** where Stuttgart RSC 1997 effective core potential with a corresponding basis set was used. The DMSO solvent was accounted for with the PCM model. The Guassian09 suite of programs (revision

D.01) was used for the calculations[60]. The excitation energies and oscillator strengths with the configuration interaction with singles (CIS) and restricted open-shell configuration interaction with singles (ROCIS) methods were calculated with the 6–31+g* basis except for the **Pd** structure, where the aug-cc-pVDZ-PP basis set[61] was used. The solvent effects were not taken into account. The ORCA v4.2.0 program was used for these calculations[62].

**Non-adiabatic dynamics.** The ground state density was sampled with molecular dynamics coupled to the Generalized Langevin Equation (GLE) thermostat set to account for nuclear quantum effects[63]. The parameters for the GLE thermostat were generated in GLE4MD.org. The propagation was performed with the Verlet method with a time step set to 20 a.u., and the energies and gradients were calculated with the orthogonalization model 3 (OM3) method. The temperature was set to 300 K. The non-adiabatic simulations were performed within the Landau–Zener surface hopping algorithm[64]. The time step for the excited state simulations was set to 5 a.u., and the energies and gradients were calculated at the multi-reference configuration interaction with singles and doubles MR-CISD/OM3(12,10) level. All dynamic simulations were performed in ABIN code[65]; semiempirical OM3 gradients and energies were calculated in the MNDO package[66].

**Spin-orbit coupling.** The Spin–orbit couplings (SOCs) were calculated at the ωB97X/def2-TZVP and DKH-ωB97X/DKH-def2-TZVP levels with a mean-field/effective potential approach, including 1-electron terms, seminumeric computation of the Coulomb term, and fully analytic computation of the exchange terms[67]. The DMSO solvent was included via PCM. For the SOC calculation, the ORCA v4.2.0 program was used[62].

**Activation energies and free energies.** The activation energies for the LG release at the complete active space self-consistent field—CASSCF(4,4)/6–31+g* and $n$-electron valence state perturbation theory—NEVPT2(4,4)/6–31+g* levels—were calculated in the ORCA v4.2.0 program[62]. The effect of solvation was accounted for through the PCM model. The optimizations and constrained scans at the BMK level were calculated with the Gassian09 suite of programs (revision D.01)[60]. The solvent was included via the PCM model.

## Data availability
The source data underlying Figs. 2b–d, f, 3b, c, e, f, 4b, c, 5b, c and Supplementary Figs 3, 6a–c, 7b–d are recorded in a Source Data file. The authors declare that other data related to this research are available within the paper and its Supplementary Information or from the authors upon request. Crystallographic data have been deposited at the Cambridge Crystallographic Data Centre (CCDC) as 1973540 and can be obtained free of charge from the CCDC via www.ccdc.cam.ac.uk/getstructures. Source data are provided with this paper.

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

## Acknowledgements

This study was supported by funding from the European Research Council (GAtransport to R.W.), the Czech Science Foundation (GA21-01799S (P.K.) and GA20-15825S (P.S.)), the JSPS KAKENHI (JP18J20500 to Y.C.), and the Global Research and Training Fellowship (GRTF) of Tel Aviv University (to A.R.S.). P.K. thanks the RECETOX Research Infrastructure (No. LM2018121) financed by the Ministry of Education, Youth and Sports, and the Operational Programme Research, Development and Education (the CETOCOEN EXCELLENCE project No. CZ.02.1.01/0.0/0.0/17_043/0009632) for supportive background. This project was also supported by the European Union's Horizon 2020 Research and Innovation Programme under grant agreement No. 857560 (P.K.). The publication reflects only the authors' view and the European Commission is not responsible for any use that may be made of the information it contains.

## Author contributions

A.R.S. and R.W. conceived the project and designed the initial experiments. A.R.S. and A.R. performed the synthesis and characterization of the metal complexes and the organic compounds and performed photophysical measurements under the supervision of R.W. and P.K. A.R.S., Y.C., and A.R. designed and performed the experimental mechanistic investigation under the supervision of P.K. J.J. designed and performed the quantum-chemical calculations under the supervision of P.S. Cellular studies were designed and performed by A.R.S. and S.I.D. under the supervision of R.S.F. The manuscript was written by A.R.S., P.K., P.S., and R.W. All authors analyzed the data, discussed the results, and commented on the manuscript.

## Competing interests

The authors declare no competing interests.
