## [Peer Review File · Nature Communications]

REVIEWER COMMENTS

Reviewer #1 (Remarks to the Author):

This manuscript describes original photoremovable protecting groups (PPGs), leading to a new class of light sensitive compound enabling a light induced concentration jump of free molecules in complex chemical or biological environments. Therefore, the authors were able to apply the well-known porphyrin chromophores for the development of a hybrid class of PPGs, which allows the incorporation of a metal ion as a part of the chromophore and releases leaving groups through the photoinduced cleavage of covalent bonds. More importantly the authors were able to demonstrate that the ability of porphyrin to chelate a large number of metal cations enables to increase the photolytic efficiency; and a 12 time increase in the uncaging efficiency was observed for a Zinc complex using a blue light excitation. The visible light sensitivity of these new classes of chromophore could potentially allow more sophisticated applications of PPG in particular in biology, especially below 630nm. However, in Fig 2d, the authors are reporting the absorbance evolution of **6** after excitations respectively at 410nm, 545 nm and 640nm. I think, it would be more accurate to present the DMPA release in order to be able to fully evaluate the potential of those PPGs. The authors were also able to nicely apply these porphyrin PPGs for the release of an antifolate agent (MTX) on cell using a 545nm excitation. I wondering why such an interesting tool was not used at higher wavelength. In addition, the author should also consider to use of Zn or Pd complexes of **10** in order to fully evaluate the potential of their PPGs on true biological applications. The mechanistic studies of these new class of PPGs are well described and documented. The Experimental parts are mostly well-described too.

In Summary, I think there is be a great potential here, in particular since the described compounds seems to be efficient using red shifted light excitation. Therefore this manuscript could be significant enough to deserve a publication in Nature communication, after major corrections.

The following specific comments should also be addressed.

- For compound **10**, since two regioisomers of the caged MTX can be formed, the authors should provide additional NMR data to confirm the structure.
- The authors should carefully check the reported ¹H NMR splitting data for compound **11**, **12** and **13**.
- In the SI figure 6a. How can compound **9** liberate 16 micomolar of Indibulin using **9** at 3 micromolar?
- In the SI figure 6f, what is the concentration of **10** and the type of solvent used in this experiment?
- In table 3: the decomposition quantum yield should $\times 10^{-3}$ not $\times 10^3$
- In my opinion, compound **19** is not bringing much to the discussion, the authors should at least add the photophysical and photochemical properties of this compound.

Reviewer #2 (Remarks to the Author):

I enjoyed reading this manuscript very much. Weinstain and Klan establish an unprecedented, new uncaging platform. It is true that there are "dozens of PPG types [which] have been developed". Still I think that this one is especially promising due to its versatility. The authors make use of the rich photochemistry of porphyrins. The fact that finetuning of the properties is possible using (or omitting) central metal ions is intriguing. The absorbance in the red-shifted part of the spectrum also distinguishes this approach from other uncaging platforms. The possibility to perform multiple uncaging out of the same precursor can have very interesting applications.

The introduction is very good. I like the comparison of the two PPG narratives (organometallic and organic) that are now optionally brought together.

The investigations are of high quality - as we are used to seeing from both labs. The quantum product obtained with many of the investigated derivatives shows that this is already a concept that can be practically used. The fact that the authors include a rich photophysical characterization as well as mechanistic studies and theoretical calculations and even first cellular experiments strengthens the value of this paper.

Do the authors have an idea why compound 9 is thermally unstable?

The tox aspects of the compounds and the uncaging byproducts are a bit undocumented. I believe it could be easy to find some explicit arguments for the main text even from the experiments that have already been done.

Figure 2 is too packed for me and it took me a long time to digest its content properly. I am not happy with the normalized absorbance in panel b as it takes away all the information of the relative epsilons.

I am interested in seeing which leaving groups are tolerated but this should be the story of another paper.

The use of superscript descriptors in Table 2 is very confusing. Maybe there is another solution that is compatible with the guidelines of the journal.

What is the reason for the delayed reaction in Figure 4a, black dots?

In summary I come to the conclusion that this is not just another PPG but could really open a new chapter in this field. The corrections I suggest are minor corrections to me. I believe that Nat. Commun. is a very good choice for this investigation and I suggest that the paper be accepted after minor editing.

Reviewer #3 (Remarks to the Author):

The authors conceive of a clever approach to PPG design that blurs the lines between two

different classes of PPG scaffolds. The authors combine organic-based PPGs with metal-complex PPGs into a hybrid of the two, where it is hoped that the identity of the metal can be used to tune photophysical properties and release efficiency. The premise of the study is certainly compelling, but the results of the strategy appear to provide a PPG that is not greater than the sum of its parts. Therefore, the significance of this work does not warrant publication in Nature Communications.

While the authors do report a novel organic PPG scaffold, a point that could be more clearly articulated in the text, the photochemical yield of release is lower than other reported PPGs in use and listed in Figure 1. This limitation may have been easy to over look if the following statement by the authors was true...

"We then show that the functional separation of the metal-binding chromophore and the site of leaving group release can be leveraged to fine-tune its spectroscopic and photochemical properties by simply introducing metal ions into the porphyrin core."

Based on the data, this reviewer deems this statement as overly generous and a case of "over promise, under deliver." The addition of metals to the core does little to appreciably change the absorption characteristics of the PPGs (Table 1 and Figure 3b). Only one example (6-Zn) even increases the quantum yield of photo-decomposition (increasing from 0.002 to 0.02). The photophysical characterization of the molecules appears rigorous and sound, but ultimately, the authors conclude that singlet oxygen generation is the major product of irradiation as opposed to release of the leaving group.

Other comments are listed below.

1) Large portions of the discussion are supported by figures that are not in the main text but referenced in the SI. This study may not be best-suited to a communication format.

2) Figure 2 is very dense and difficult to read. Consider breaking it up with the discussion and making the individual elements larger. Figure 1 was highly effective!

3) There is an inconsistency in representing the quantum yields of decomposition and singlet oxygen generation. One is represented as a percent. They should be kept uniform in representation.

4) The NMR spectra of newly synthesized compounds is not provided in the SI.

Response to reviewers' comments

Reviewers comments

Reviewer 1

This manuscript describes original photoremovable protecting groups (PPGs), leading to a new class of light sensitive compound enabling a light induced concentration jump of free molecules in complex chemical or biological environments. Therefore, the authors were able to apply the well-known porphyrin chromophores for the development of a hybrid class of PPGs, which allows the incorporation of a metal ion as a part of the chromophore and releases leaving groups through the photoinduced cleavage of covalent bonds. More importantly the authors were able to demonstrate that the ability of porphyrin to chelate a large number of metal cations enables to increase the photolytic efficiency; and a 12 time increase in the uncaging efficiency was observed for a Zinc complex using a blue light excitation. The visible light sensitivity of these new classes of chromophore could potentially allow more sophisticated applications of PPG in particular in biology, especially below 630 nm. However, in Fig 2d, the authors are reporting the absorbance evolution of **6** after excitations respectively at 410 nm, 545 nm and 640 nm. I think, it would be more accurate to present the DMPA release in order to be able to fully evaluate the potential of those PPGs.

The release of DMAP following photoexcitation of **6** at the different wavelengths is now shown in Figure 2d, instead of its photodecomposition.

The authors were also able to nicely apply these porphyrin PPGs for the release of an antifolate agent (MTX) on cell using a 545 nm excitation. I wondering why such an interesting tool was not used at higher wavelength.

To demonstrate irradiation at >545 nm, we performed the photoactivation of compound **13** (updated compound number from **10**) in cultured cells using 640 nm light. Compound **5**, releasing the nontoxic *p*-nitrobenzoic acid served as a negative control. The experiment was repeated three times in triplicates (the complete experimental details are reported in the SI). The results show that photoactivation of **13** using 640 nm light led to an 8-fold decrease in IC₅₀ (from ~24 to ~3 μM), a value comparable to free MTX at a similar concentration. The irradiation time (10 min) was double than that required to achieve the same effect using 545 nm light (5 min). These results verify that 640 nm light is sufficient to promote effective photoactivation of the novel PPG reported herein in relevant biological settings.

The experiment is now mentioned in the main text and the data are shown in Figure 3f and in Supplementary Figure 7.

In addition, the author should also consider to use of Zn or Pd complexes of **10** in order to fully evaluate the potential of their PPGs on true biological applications.

In this manuscript, we focus on introducing and examining the concept of organic/metal-complex hybrid PPGs, using *meso*-methylporphyrin as a prototype. We performed a comprehensive investigation of the porphyrin scaffold as a novel PPG (structure, mechanism and a potential bio-application), in which the metal-complex derivatives serve as extended

examples. We currently study the photochemistry of a wide array of metallo-porphyrin derivatives and their biological applications in our lab. This is an extension of the project that still requires a lot of experimental data and therefore, we plan to describe these results in a subsequent manuscript.

The mechanistic studies of these new class of PPGs are well described and documented. The Experimental parts are mostly well-described too.

In Summary, I think there is be a great potential here, in particular since the described compounds seems to be efficient using red shifted light excitation. Therefore this manuscript could be significant enough to deserve a publication in Nature communication, after major corrections.

The following specific comments should also be addressed.

- For compound **10**, since two regioisomers of the caged MTX can be formed, the authors should provide additional NMR data to confirm the structure.

HPLC and $^1\text{H}/^{13}\text{C}$ -NMR show the presence of a single isomer of **13** (updated compound number from **10**). Based on correlation NMRs (COSY, HSQC, figures 1 and 2, at the end of this document), we identified individual protons signals in the ^1H -NMR spectrum (Figure 3). We then identified the signals of the hydrogens and carbons that are part of, and around, the newly formed ester bond connecting the porphyrin to MTX (Figure 4). It should be noted that in the ^{13}C -NMR spectrum, the sp^2 pyrrolic carbons of porphyrin are not shown distinctly, and the carbonyl carbons appear with a small intensity after 10,000 scans. We then analyzed long-distance C-H correlations using HMBC (Figure 5). On careful analysis of long-range C-H correlation (HMBC) NMR of **13**, we found an interaction between the methylene hydrogens of porphyrin (H1) with the carbonyl carbon C2 (blue arrowed interaction). The C2 carbon also interacts with H3, verifying its identity. We did not find any evidence for interaction between the methylene hydrogens of porphyrin (H1) and the carbonyl carbon C6 in methotrexate. These observations establish the proposed structure of **13** as the correct isomer.

Our finding is in accord with previous literature showing a higher reactivity of the γ -COOH to such coupling reactions.¹⁻³

All these data are now included in the SI (Supplementary Figures 67-72).

- The authors should carefully check the reported ^1H NMR splitting data for compound **11**, **12** and **13**.

We thank the reviewer for this comment. The reported ^1H -NMR splitting data for compounds **9-11** (updated compound number for **11-13**) was indeed reported erroneously. We corrected our report in the SI.

We were trying to convey in our report the fact that the compounds show an additional, minor set of peaks for the amino acid(s) hydrogens. This is a result of the rotation barrier in Boc-Trp that leads to the formation of rotamers.⁴⁻⁶ The ^1H -NMR spectrum of a commercially obtained Boc-Trp-OH exhibits two singlet peaks (rotamers) for the *tert*-butyl protons at 1.336 and 1.217 ppm, with the combined integration of 9 protons (7.7+1.7) at room temperature (see figure 6 at the end of this document). Therefore, we added HPLC traces for compounds **11-13**, demonstrating they elute as single peaks. In addition, we provide VT-NMR spectra

(25 to 80 °C) (Figures 7-9 at the end of this document), showing the merging of the double set of peaks into a single set, supporting the notion that they emanate from rotamers. These data were added to the relevant SI section (Supplementary Figures 56-64).

- In the SI figure 6a. How can compound **9** liberate 16 micromolar of Indibulin using **9** at 3 micromolar?

The concentration stated (3 μM) for compound **12** (updated compound number from **9**) was incorrect. The correct concentration is 25 μM (used for all HPLC experiments). This figure was moved to the main text as Figure 3c with the corrected figure caption.

- In the SI figure 6f, what is the concentration of **10** and the type of solvent used in this experiment?

Compound **13** (updated compound number from **10**) was used at 25 μM in DMSO. This figure was moved to the main text as Figure 3c, and the additional information is now included in the figure caption.

- In table 3: the decomposition quantum yield should $\times 10^{-3}$ not $\times 10^3$.

The values of decomposition quantum yields shown in Table 1 were presented as $\Phi_{dec} \times 10^3$. For example, a value of 1.7 then represents a decomposition quantum yield of 0.0017.

Nevertheless, to avoid confusion, we now present the data as decimal numbers (0.0017).

- In my opinion, compound **19** is not bringing much to the discussion, the authors should at least add the photophysical and photochemical properties of this compound.

Indeed, the compound does not provide additional information on the photochemistry, as we expect that it has similar properties to its analogous derivatives. This is why its photochemistry was not studied in detail. On the other hand, the compound demonstrates that copper-mediate click works on *meso*-methylporphyrin, without the exchange of the chelated metal ion. In our view, it highlights an important structural point by establishing a convenient route for further functionalization of the PPG scaffold, for example by conjugating it to polymers, targeting motifs (such as antibodies) or water-solubilizing moieties.

Reviewer 2

I enjoyed reading this manuscript very much. Weinstain and Klan establish an unprecedented, new uncaging platform. It is true that there are "dozens of PPG types [which] have been developed". Still I think that this one is especially promising due to its versatility. The authors make use of the rich photochemistry of porphyrins. The fact that finetuning of the properties is possible using (or omitting) central metal ions is intriguing. The absorbance in the red-shifted part of the spectrum also distinguishes this approach from other uncaging platforms. The possibility to perform multiple uncaging out of the same precursor can have very interesting applications.

The introduction is very good. I like the comparison of the two PPG narratives (organometallic and organic) that are now optionally brought together.

The investigations are of high quality - as we are used to seeing from both labs. The quantum product obtained with many of the investigated derivatives shows that this is already a concept that can be practically used. The fact that the authors include a rich photophysical characterization as well as mechanistic studies and theoretical calculations and even first cellular experiments strengthens the value of this paper.

Do the authors have an idea why compound **9** is thermally unstable?

Compound **12** (updated compound number for **9**) decomposes in 1:1 DMSO:PBS buffer (pH 7.4) in the dark to release free indibulin (see figure 10 at the end of this document). Previous studies have demonstrated the susceptibility of the “benzylic” position in porphyrin to nucleophilic attack when bearing a good leaving group, and specifically, a quaternary ammonium salt.^{7, 8} We presume that an excellent leaving group (indibulin) contribute to high reactivity of the *meso*-methyl position to nucleophilic attack by water molecules. In comparison to DMAP as a leaving group, the major difference is the fact that the pyridinium group of indibulin bears considerably less electron-donating oxoacetamido group (DMAP). This makes the group a much better LG and its release in the dark.

The tox aspects of the compounds and the uncaging byproducts are a bit undocumented. I believe it could be easy to find some explicit arguments for the main text even from the experiments that have already been done.

We provided cellular toxicity data (using Coulter counter) for compounds **5** and **13** in the dark and following irradiation with 545 nm light. We now added data on their toxicity following irradiation with 640 nm light, as well as a comparison to that of free MTX. We did not test toxicity aspects of photoproducts because our data show that the photoexcitation of **6** leads mostly to unidentified photodecomposition products (Supplementary Figure 9). Although we have data suggesting that the light-dependent toxicity of **13** is a combination of the effects of the drug released and singlet oxygen generation, we believe that more data is needed to fully understand the process and therefore refrain from making any assertions in the text.

Figure 2 is too packed for me and it took me a long time to digest its content properly.

We thank for the suggestion. The original figure was split into two separate figures: Figures 2 and 3. Figure 2 now presents metal-free derivatives of the *meso*-methylporphyrin, including spectroscopic and photochemical properties. Figure 3 focuses on drug-bearing derivatives and their evaluation in cultured cells.

I am not happy with the normalized absorbance in panel b as it takes away all the information of the relative epsilons.

Panel 2b and 3b (normalized absorbance) was replaced to show relative absorbance.

I am interested in seeing which leaving groups are tolerated but this should be the story of another paper.

Indeed. We expect a wide range eaving groups to be tolerated. The challenge at the moment seems to be on the synthetic side, as several common transformations of the *meso*-methylalcohol moiety were unsuccessful (mentioned in the manuscript).

The use of superscript descriptors in Table 2 is very confusing. Maybe there is another solution that is compatible with the guidelines of the journal.

We replaced all superscript descriptors with symbols in Table 2 to eliminate the confusion.

What is the reason for the delayed reaction in Figure 4a, black dots?

Our results suggest that initial $^1\text{O}_2$ generation by the excited porphyrin outcompetes the photorelease, in accord with the measured quantum yields of both processes (Φ_{Δ} is ~2 orders of magnitude higher than Φ_{dec}). Once ground state oxygen is partially depleted from the immediate environment, the photorelease can proceed efficiently. Thus, the unusual course of photochemistry observed in aerated DMSO solutions most probably reflects the initial quenching of the triplet excited state to form singlet oxygen, which reacts with DMSO molecules;⁹ this does not occur in methanol. A similar effect of local molecular oxygen depletion on a photoreaction has recently been reported for the photoactivation of gold(I) arylethynyl complexes phosphorescence.¹⁰

In summary I come to the conclusion that this is not just another PPG but could really open a new chapter in this field. The corrections I suggest are minor corrections to me. I believe that Nat. Commun. is a very good choice for this investigation and I suggest that the paper be accepted after minor editing.

Reviewer 3

The authors conceive of a clever approach to PPG design that blurs the lines between two different classes of PPG scaffolds. The authors combine organic-based PPGs with metal-complex PPGs into a hybrid of the two, where it is hoped that the identity of the metal can be used to tune photophysical properties and release efficiency.

The premise of the study is certainly compelling, but the results of the strategy appear to provide a PPG that is not greater than the sum of its parts. Therefore, the significance of this work does not warrant publication in Nature Communications.

We would like to clarify that the newly introduced PPG is not a combination of two (or more) known PPGs. It is a single entity that could be manipulated through several structural modifications, especially through one unprecedented: metal chelation. It should not be considered as the combination/amalgamation of parts in which each contributes its own properties but as a novel type of a moiety possessing unique properties.

While the authors do report a novel organic PPG scaffold, a point that could be more clearly articulated in the text, the photochemical yield of release is lower than other reported PPGs in use and listed in Figure 1.

If the reviewer means quantum yield, the value itself is not relevant when considering PPGs absorbing wavelengths in the whole UV/VIS/NIR spectrum. The reason is considerably different molar absorption coefficients spanning many orders of magnitude. Instead, uncaging

cross section ($\Phi_r \varepsilon(\lambda_{\text{irr}})$) is the most relevant value for any application of a PPG (just like “brightness”, i.e. $\Phi_F \varepsilon(\lambda_{\text{irr}})$, is the most relevant term for comparing fluorescent molecules). When considering PPGs in Figure 1 and other relevant systems, our porphyrin derivatives is by far the most versatile and flexible PPG ever reported, and thanks to its high cross sections throughout a significant part of the visible spectrum, none of any individual PPG in Figure 1 can provide this even remotely.

This limitation may have been easy to overlook if the following statement by the authors was true... “We then show that the functional separation of the metal-binding chromophore and the site of leaving group release can be leveraged to fine-tune its spectroscopic and photochemical properties by simply introducing metal ions into the porphyrin core.” Based on the data, this reviewer deems this statement as overly generous and a case of “over promise, under deliver.” The addition of metals to the core does little to appreciably change the absorption characteristics of the PPGs (Table 1 and Figure 3b). Only one example (6-Zn) even increases the quantum yield of photo-decomposition (increasing from 0.002 to 0.02).

We respectfully beg the difference on this point. We consider **one order of magnitude enhancement** in the quantum yield as one of the major achievements in our manuscript. Such an enhancement usually requires a very long and tedious research on any individual PPG chromophore modification.^{11, 12} In our case, the exchange of a core metal ion is straightforward and easily done. In addition, since porphyrin has the capacity to chelate many metal ions that were not evaluated here, the scope for further gains is still broad.

The photophysical characterization of the molecules appears rigorous and sound, but ultimately, the authors conclude that singlet oxygen generation is the major product of irradiation as opposed to release of the leaving group.

Whether this property is a limitation or an advantage is strictly dependent on the intended application. Some major works consider this an advantage, especially when anti-cancer substances are released.¹³ It is important to note, though, that $^1\text{O}_2$ does not destroy the PPG during irradiation. In addition, several mitigation strategies for $^1\text{O}_2$ generation by porphyrins are known¹⁴⁻¹⁸ and could potentially be implemented in this case, although whether this can be done without affecting photorelease is remained to be tested (as we discuss in our conclusions).

Other comments are listed below.

Large portions of the discussion are supported by figures that are not in the main text but referenced in the SI. This study may not be best-suited to a communication format.

We re-arranged the figures in the manuscript and added a new figure that summarizes the main findings deliberated in the discussion (new Figure 6). We also added references to the specific supplementary figures and tables that support each conclusion in this section.

Figure 2 is very dense and difficult to read. Consider breaking it up with the discussion and making the individual elements larger. Figure 1 was highly effective!

According to the suggestion, the original figure was split into two separate figures: Figures 2 and 3. Figure 2 now presents the metal-free derivatives of *meso*-methylporphyrin, including spectroscopic and photochemical properties. Figure 3 focuses on drug-bearing derivatives and their evaluation in cultured cells.

There is an inconsistency in representing the quantum yields of decomposition and singlet oxygen generation. One is represented as a percent. They should be kept uniform in representation.

We eliminated the use of percents for quantum yields of decomposition or singlet oxygen generation throughout the manuscript.

The NMR spectra of newly synthesized compounds is not provided in the SI.

All newly synthesized compounds were comprehensively characterized, and the data were included in the SI. We added additional NMR data for compounds **10-13** (addressing specific questions of reviewers 1 and 2), and all is now added to the SI. Photoproducts **14**, **20** and **21**, formed in trace amounts, were characterized by HPLC-MS analyses.

1. Fan, Z.; Wang, Y.; Xiang, S.; Zuo, W.; Huang, D.; Jiang, B.; Sun, H.; Yin, W.; Xie, L.; Hou, Z., Dual-self-recognizing, stimulus-responsive and carrier-free methotrexate–mannose conjugate nanoparticles with highly synergistic chemotherapeutic effects. *Journal of Materials Chemistry B* **2020**, *8* (9), 1922-1934.
2. Rosowsky, A.; Forsch, R. A.; Yu, C. S.; Lazarus, H.; Beardsley, G. P., Methotrexate analogs. 21. Divergent influence of alkyl chain length on the dihydrofolate reductase affinity and cytotoxicity of methotrexate monoesters. *J. Med. Chem.* **1984**, *27* (5), 605-609.
3. Hou, M.; Li, S.; Xu, Z.; Li, B., A Reduction-responsive Amphiphilic Methotrexate-Podophyllotoxin Conjugate for Targeted Chemotherapy. *Chemistry – An Asian Journal* **2019**, *14* (21), 3840-3844.
4. Donohoe, T. J.; Cheeseman, M. D.; O'Riordan, T. J. C.; Kershaw, J. A., Synthesis of (+)-DGDP and (–)-7-epialexine. *Organic & Biomolecular Chemistry* **2008**, *6* (21), 3896-3898.
5. Johnston, C. P.; Smith, R. T.; Allmendinger, S.; MacMillan, D. W. C., Metallaphotoredox-catalysed sp³–sp³ cross-coupling of carboxylic acids with alkyl halides. *Nature* **2016**, *536* (7616), 322-325.
6. Giuliano, M. W.; Maynard, S. J.; Almeida, A. M.; Reidenbach, A. G.; Guo, L.; Ulrich, E. C.; Guzei, I. A.; Gellman, S. H., Evaluation of a Cyclopentane-Based γ -Amino Acid for the Ability to Promote α/γ -Peptide Secondary Structure. *The Journal of Organic Chemistry* **2013**, *78* (24), 12351-12361.
7. Yashunsky, D. V.; Ponomarev, G. V.; Arnold, D. P., Chemistry of dimethylaminomethylporphyrins. 2. Porphyrin dimers linked by pyrrolylmethylene units. *Tetrahedron Lett.* **1997**, *38* (1), 105-108.
8. Yashunsky, D. V.; Arnold, D. P.; Ponomarev, G. V., Porphyrins. 37. Synthesis of heterodimers of porphyrins and chlorins containing pyrrolyl-methyl bridges. *Chemistry of Heterocyclic Compounds* **2000**, *36* (3), 275-280.
9. Lutkus, L. V.; Rickenbach, S. S.; McCormick, T. M., Singlet oxygen quantum yields determined by oxygen consumption. *Journal of Photochemistry and Photobiology A* **2019**, *378*, 131-135.

10. Wan, S.; Lu, W., Reversible Photoactivated Phosphorescence of Gold(I) Arylethynyl Complexes in Aerated DMSO Solutions and Gels. *Angew. Chem. Int. Ed.* **2017**, *56* (7), 1784-1788.
11. Klán, P.; Šolomek, T.; Bochet, C. G.; Blanc, A.; Givens, R.; Rubina, M.; Popik, V.; Kostikov, A.; Wirz, J., Photoremovable Protecting Groups in Chemistry and Biology: Reaction Mechanisms and Efficacy. *Chem. Rev.* **2013**, *113* (1), 119-191.
12. Weinstain, R.; Slanina, T.; Kand, D.; Klán, P., Visible-to-NIR-Light Activated Release: From Small Molecules to Nanomaterials. *Chem. Rev.* **2020**, *120* (24), 13135-13272.
13. Luo, D.; Carter, K. A.; Miranda, D.; Lovell, J. F., Chemophototherapy: An Emerging Treatment Option for Solid Tumors. *Advanced Science* **2017**, *4* (1), 1600106.
14. Hirakawa, K.; Harada, M.; Okazaki, S.; Nosaka, Y., Controlled generation of singlet oxygen by a water-soluble meso-pyrenylporphyrin photosensitizer through interaction with DNA. *Chem. Commun.* **2012**, *48* (39), 4770-4772.
15. Hirakawa, K.; Nishimura, Y.; Arai, T.; Okazaki, S., Singlet Oxygen Generating Activity of an Electron Donor Connecting Porphyrin Photosensitizer Can Be Controlled by DNA. *J. Phys. Chem. B* **2013**, *117* (43), 13490-13496.
16. Jeong, H.-G.; Choi, M.-S., Design and Properties of Porphyrin-based Singlet Oxygen Generator. *Isr. J. Chem.* **2016**, *56* (2-3), 110-118.
17. Mathai, S.; Smith, T. A.; Ghiggino, K. P., Singlet oxygen quantum yields of potential porphyrin-based photosensitisers for photodynamic therapy. *Photochem. Photobiol. Sci.* **2007**, *6* (9), 995-1002.
18. McCarthy, J. R.; Weissleder, R., Model Systems for Fluorescence and Singlet Oxygen Quenching by Metalloporphyrins. *ChemMedChem* **2007**, *2* (3), 360-365.

Figure 1. COSY NMR spectrum of **13** (updated compound number for**10**) in DMSO-d₆.

Figure 2. HSQC NMR spectrum of **13** (updated compound number for**10**) in DMSO-d₆.

Figure 3. Analyzed $^1\text{H-NMR}$ spectrum and structure of **13** (updated compound number for **10**) in $\text{DMSO-}d_6$ at room temperature.

Figure 4. Relevant peaks on ^1H - (top) and ^{13}C -NMR (bottom) spectra of **13** (updated compound number for **10**) in DMSO-d_6 at room temperature.

Figure 5. Long-range C-H correlations in HMBC NMR spectrum of **13** (updated compound number for **10**). *Top*: identified interactions. The interactions within the MTX structure are marked in red. The key interaction H1-C2 is marked in blue. *Bottom*: analyzed HMBC NMR spectrum of **10**. Shown is the section relevant for structure determination.

Figure 6. $^1\text{H-NMR}$ spectrum of Boc-Trp-OH in DMSO-d_6 . The arrows mark the two peaks of the Boc hydrogens emanating from the two rotamers existing at room temperature.

Figure 7. (a) HPLC traces and (b) VT-NMR of **9** (updated compound number for **11**) in DMSO- d_6 at 25 to 80 °C.

Figure 8. (a) HPLC traces and (b) VT-NMR of **10** (updated compound number for **12**) in DMSO- d_6 at 25 to 80 °C.

Figure 9. (a) HPLC traces and (b) VT-NMR of **11** (updated compound number for **13**) in DMSO-d₆ at 25 to 80 °C.

Figure 10. Decomposition of compound **12** (updated compound number for **9**) in the dark. Compound **12** (25 μ M) was dissolved in 1:1 DMSO:PBS buffer (pH 7.4) and kept in the dark at room temperature. Samples were analyzed by HPLC-MS against authentic standard (compound **12**, indibulin) at times 0, 1 and 2 h after incubation.

REVIEWERS' COMMENTS

Reviewer #1 (Remarks to the Author):

The authors have largely revised the manuscript. They properly addressed all concerns in my previous report. I can now recommend publication in Nature Communications.

Reviewer #2 (Remarks to the Author):

In this revised version, the authors have addressed all my concerns very well. I also think that the same holds true for the concerns of the other reviewers - especially the ones of the most critical reviewer 3. Along with the response of the authors, I also do not see at all why one should be able to argue that this paper is incremental - especially since reviewer 3 states himself that we are looking at a "novel organic PPG scaffold" here. I fully support publishing in Nat. Commun. and expect that this paper will make quite an impact.

Reviewer #3 (Remarks to the Author):

The changes made to the manuscript help it along its way to being a strong article. The work combines organic and metal free PPG's through generation and demonstration of a new class of porphyrin PPGs. The only remaining concern might be that the manuscript reads like a data dump at times, detracting from the uniqueness of the concept. A lot of interesting data is relegated to the SI, which is a shame. This is provided as a critique for the authors and editor to consider (article versus comm.) and not on the merits of the science.

Response to Reviewers

Reviewer #1 (Remarks to the Author):

The authors have largely revised the manuscript. They properly addressed all concerns in my previous report. I can now recommend publication in Nature Communications.

Thank you for helping us improve the manuscript.

Reviewer #2 (Remarks to the Author):

In this revised version, the authors have addressed all my concerns very well. I also think that the same holds true for the concerns of the other reviewers - especially the ones of the most critical reviewer 3. Along with the response of the authors, I also do not see at all why one should be able to argue that this paper is incremental - especially since reviewer 3 states himself that we are looking at a “novel organic PPG scaffold” here. I fully support publishing in Nat. Commun. and expect that this paper will make quite an impact.

Thank you for helping us improve the manuscript.

Reviewer #3 (Remarks to the Author):

The changes made to the manuscript help it along its way to being a strong article. The work combines organic and metal free PPG's through generation and demonstration of a new class of porphyrin PPGs. The only remaining concern might be that the manuscript reads like a data dump at times, detracting from the uniqueness of the concept. A lot of interesting data is relegated to the SI, which is a shame. This is provided as a critique for the authors and editor to consider (article versus comm.) and not on the merits of the science.

Thank you for helping us improve the manuscript. We agree that a significant amount of relevant data is only included in the SI however, all of it is referenced directly in the main text and we trust that the readers will find it easy to navigate between the two.